# Long-Term Spatiotemporal Variation and Environmental Driving Forces Analyses of Algal Blooms in Taihu Lake Based on Multi-Source Satellite and Land Observations

**Tiantian Zhang** [1,2]**, Hong Hu** [1,2],*** , Xiaoshuang Ma** [1,2] **and Yaobo Zhang** [3]

1   School of Resources and Environmental Engineering, Anhui University, Hefei 230601, China;
    candy15665330929@163.com (T.Z.); mxs.88@163.com (X.M.)
2   Anhui Provincial Key Laboratory of Wetland Ecological Protection and Restoration, Hefei 230601, China
3   Anhui Provincial Bureau of Surveying and Mapping, Hefei 230601, China; yaobozh@msn.com
*   Correspondence: huhong@ahu.edu.cn; Tel.: +86-152-5607-5296

**Abstract:** The algal blooms caused by the eutrophication of lakes is a major environmental problem. In this study, we took China's Taihu Lake as the research area, using multi-source satellite imagery data to monitor the information of algal blooms from 2008 to 2017. Following the analyses of the temporal and spatial variation trends of the blooms, water quality and meteorological data from land observation stations were employed to investigate the main environmental driving forces of the algal bloom outbreaks. The results show that, over the decade, the blooms with medium and higher hazard degrees mainly occurred in summer and autumn, and especially in autumn. From 2008 to 2016, the algal blooms outbreak degree was relatively stable, but, in 2017, it was severe, and the Northwest Lake area and the northern bays had heavier blooms than the other lake areas. From the analyses of the environmental driving forces, the variation trend of total nitrogen (TN) and total phosphorus (TP) concentrations in Taihu Lake from 2008 to 2017 was moderate, and the minimum concentrations of TN and TP both exceeded the threshold for algal bloom outbreaks. It was also found that the algal bloom area had notable correlations with the sunshine duration, wind speed and direction, precipitation, and air pressure. The research results of this paper will provide a theoretical basis for the scientific prediction of the occurrence of algal blooms in Taihu Lake.

**Keywords:** algal blooms; remote sensing monitoring; spatiotemporal distribution; environmental driving force analyses

## 1. Introduction

Algal blooms are one of the most serious ecological and environmental problems affecting water bodies, and they can have a major impact on both human and ecosystem health. Taihu Lake, which is the study area of this paper, is located in the lower reaches of the Yangtze River Basin in China, which is an area with the most developed economy and densest population [1]. Taihu Lake is a typical large-scale shallow lake with various functions, including flood control, water supply, shipping, and aquaculture. The quality of its water environment is directly related to the social and economic development of the surrounding cities. Under the background of accelerated industrial and agricultural modernization and urbanization, a large amount of nutrient loading (nitrogen and phosphorus) has been input into the lake, leading to the intensified eutrophication of Taihu Lake and the outbreak of algal blooms [2,3]. In 2007, the outbreak of algal blooms in Taihu Lake led to a drinking water crisis in the city of Wuxi [4]. At the time of writing, the algal blooms are still serious, and the lake

environmental management is facing tremendous pressure [5]. Algal blooms not only decrease the water quality and damage the ecosystem structure, but they also pose a serious ecological risk through direct environmental pollution [6,7]. Therefore, investigating the temporal and spatial distribution and the main environmental driving forces of the blooms in Taihu Lake is essential for a scientific study of the ecological risks and the causes of the abnormal growth of blooms.

Field investigation by ship survey and laboratory measurements are the basis of the conventional monitoring approaches [8]. However, the field investigation only samples local point data, which cannot be used to study the dynamic spatial distribution of algal blooms over a large scale. Furthermore, this approach is expensive and labor-intensive [9,10]. Satellite remote sensing technology can provide large-scale, multi-spectral, and multi-platform algal bloom information [11], which can dynamically reflect its spatiotemporal characteristics. The satellite remote sensing approach can effectively make up for the deficiency of the conventional monitoring methods, and has become an indispensable technical means for the dynamic monitoring of the water environment, and can even provide us with an early warning of algal blooms [12].

Many scholars have explored the use of remote sensing data to monitor and study algal blooms. Hu et al. [13] used Moderate Resolution Imaging Spectroradiometer (MODIS) satellite remote sensing image data from 2000 to 2008 to identify the algal bloom information of Taihu Lake by employing the floating algae index (FAI), and analyzed its temporal and spatial distribution characteristics. Prangsma and Roozekrans [14] successfully carried out remote sensing identification and time-series analyses of algal blooms in clear water near the sea by using the relatively high reflectivity information of the Advanced Very-High-Resolution Radiometer (AVHRR) instrument. Bresciani et al. [15] monitored the bloom in the Curonian Lagoon of Lithuania with Medium Resolution Imaging Spectrometer (MERIS) and Advanced Synthetic Aperture Radar (ASAR) remote sensing data. Sòria-Perpinyà et al. [16] extracted the bloom information for a hypertrophic lagoon based on Sentinel-2A satellite data. Vincent et al. [17] developed an algorithm to measure the relative phycocyanin content (PC) and the turbidity of the western basin of Lake Erie in North America using Landsat data and revealed the temporal and spatial dynamics of the algal blooms in Lake Erie and other systems. Lastly, Wang et al. [18] used single-polarization radar data to identify blooms in Taihu Lake under rainy conditions with the support vector machine method, which provided a reference for the radar remote sensing monitoring of algal blooms in severe weather conditions.

The aforementioned studies attempted to use different remote sensing data to explore algal blooms from different aspects. Although these studies can provide a theoretical reference for algal bloom monitoring, they still have some shortcomings:

1. Temporal span and resolution: the previous studies have mostly concentrated on monitoring and analyzing algal blooms by the use of short-term datasets, which, clearly, cannot reflect the temporal and spatial distribution trends of bloom outbreaks over the long term. In addition, it should be pointed out that, unlike some other environmental problems, such as land desertification and soil erosion, which do not fluctuate significantly over a short period of time, algal blooms can evolve significantly over seasons, months, and even weeks. Thus, to obtain objective and accurate changes of the algal blooms, high temporal resolution remote sensing data are essential.

2. Data sources: most of the existing studies have focused on optical remote sensing monitoring, which is impossible in rainy and cloudy weather, making it difficult to accurately capture the information of algal bloom changes on those days. As a good supplement, radar remote sensing is not affected by cloudy weather conditions and can observe the targets both day and night. Furthermore, as previously mentioned, high temporal resolution data are essential when monitoring algal blooms, leading to the need to integrate multi-source optical and radar remote sensing data.

In this study, based on the above analysis, we utilized multi-source optical and radar remote sensing datasets, including Landsat 5–8, Gaofen-1 (GF-1), Sentinel-2, and Sentinel-1 satellite data, to identify and extract the algal bloom information for Taihu Lake from 2008 to 2017. On this basis,

the long-term temporal and spatial distributions of the algal blooms were analyzed, and the water quality and meteorological data were combined to inspect the main environmental driving forces of the blooms. The main innovations of this study are as follows:

1.  Long-term and multi-source optical and radar remote sensing data were utilized in this study to extract the algal bloom information. In addition, spatiotemporal data fusion (STF) technology [19] was employed to generate high-resolution optical images as supplementary data. This ensured that, not only could the variation characteristics of the blooms over a large time scale be objectively revealed, but we could also accurately reflect the fluctuation of algal blooms over a short time period.

2.  On the basis of analyzing the spatial and temporal distribution characteristics of the blooms with satellite data, we further analyzed the main factors affecting the blooms with water quality and meteorological data from ground stations. Such comprehensive analyses could better reveal the causes and trends of the occurrence and evolution of algal blooms.

## 2. Materials and Methods

### 2.1. Study Area

The Taihu Lake Basin is located in the lower reaches of the Yangtze River (Figure 1a). As the storage center for water resources in the basin, Taihu Lake is the third largest freshwater lake in China, with a water area of 2338 km$^2$ and an average water depth of 1.9 m. The climate of Taihu Lake is a subtropical monsoon climate with four distinct seasons. The lake spans the two provinces of Jiangsu and Zhejiang, and is responsible for urban and rural water supply in the cities of Wuxi and Suzhou, with a service scope of more than 20 million people. The large population, high urbanization level, and developed industries in the region have intensified the water pollution in Taihu Lake, causing serious outbreaks of algal blooms. Normally, Taihu Lake can be divided into eight segments (Figure 1b), namely: Central Lake, Northwest Lake, Southwest Lake, East Lake, Zhushan Bay, Meiliang Bay, Gong Bay, and East Bay [13]. However, it should be pointed out that the landscape of East Bay experienced many changes during the study period, which affected the information statistics. Therefore, East Bay was excluded from our study.

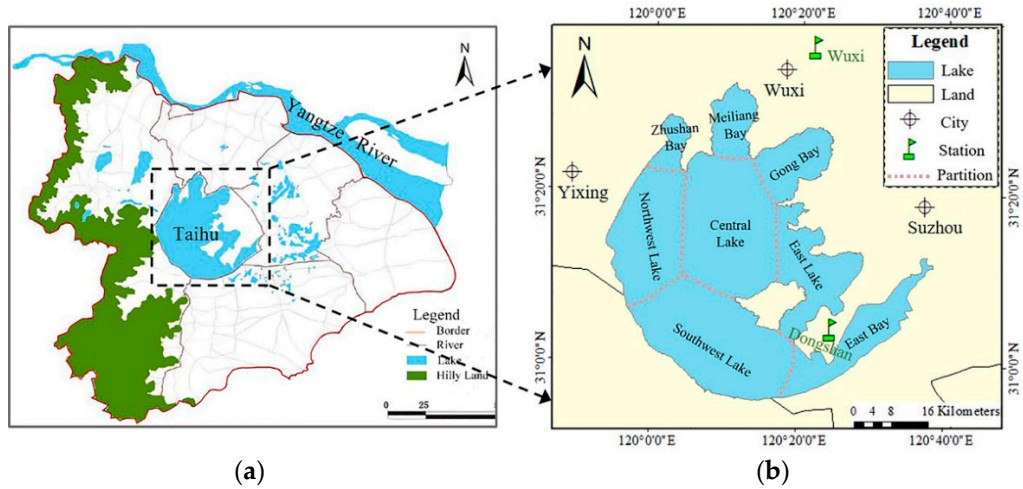

(**a**)                                          (**b**)

**Figure 1.** (**a**) Location of Taihu Lake in China. (**b**) Map of the study area.

### 2.2. Data Sources and Preprocessing

#### 2.2.1. Data Sources

Multi-source satellite imagery data, water quality indicator, and meteorological station datasets from 2008 to 2017 were used in the study. Among the data sources, the multi-source optical and

radar remote sensing data were mainly used to identify and extract the algal bloom information, and the water quality indicators and meteorological station data were used to analyze the main environmental driving forces of the bloom outbreaks. In this study, to objectively reveal the spatial and temporal distribution characteristics of the algal blooms, we collected multi-source satellite datasets, to ensure that we had at least seven high-quality images for each season. To do so, Landsat 5–8, GF-1, Sentinel-1, and Sentinel-2 remote sensing datasets were utilized. However, among the data sources, the GF-1, Sentinel-1, and Sentinel-2 satellites were launched in the second five years of the study period. Together with the influence of rainy or cloudy weather during the study period, this made the available high-quality datasets insufficient, especially for the first five years.

To deal with the above issue, STF technology was employed to generate high-quality optical images, with the help of MODIS datasets. The main processes of STF consist of constructing the relationship between the data acquired by two different sensors, and using one type of data at the target time to inverse the other type of data at the corresponding time. The details of the STF procedure are provided in Section 2.3, and the details of the utilized remote sensing datasets are given in Table 1.

**Table 1.** Details of the satellite remote sensing data used in this study.

| Product | Landsat 5–8 | GF-1 | Sentinel-2 | Sentinel-1 | MOD09GA |
|---|---|---|---|---|---|
| Spatial resolution | 30 m | 16 m | 10 m | 9–10 m | 250/500 m |
| Revisit | 16 days | 4 days | 10 days | 12 days | daily |
| Data type | Optical | Optical | Optical | Radar | Optical |
| Number | 111 | 57 | 12 | 9 | 53 |
| Time span | 2008–2017 | 2013–2017 | 2016–2017 | 2015–2016 | 2008–2017 |

### 2.2.2. Satellite Data Preprocessing

The imaging process of satellite imagery can be affected by a variety of factors, such as a change of satellite speed, the interaction between electromagnetic waves and the atmosphere, and random noise, resulting in radiometric and geometric distortion of the imagery. Therefore, preprocessing steps for the satellite remote sensing images are needed, including radiometric calibration, atmospheric correction, and geometric correction. A detailed flowchart of the preprocessing operations is given in Figure 2, where '①' means that the pretreatment was only applied to the GF-1 data, and '②' means that this preprocessing operation was only applicable to the Landsat-7 data.

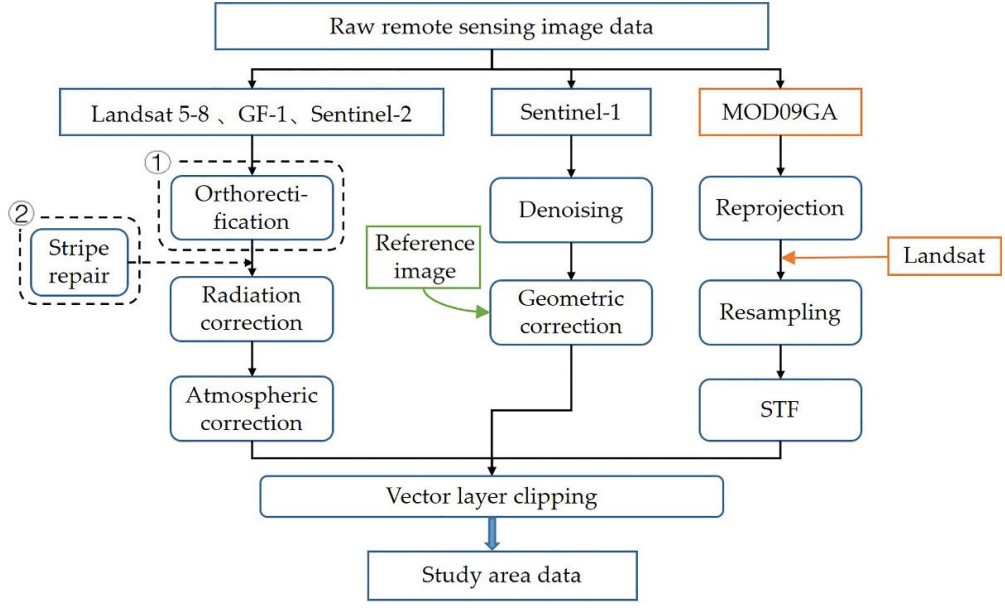

**Figure 2.** Preprocessing flowchart for the multiple remote sensing images.

## 2.3. Spatiotemporal Data Fusion

High-quality images might not be available in a certain month due to the influence of continuous cloudy weather and the long revisit period of the sensors. Therefore, we used the STF method to generate high-resolution optical images of the corresponding months. As shown in Figure 3, the basic framework of STF is to establish the relationship between the data acquired by two different sensors at close time phases, where one has a high spatial resolution but a low temporal resolution, while the other has a low spatial resolution but a high temporal resolution. We then inverse the data with a high spatial resolution using the high-temporal-resolution data of the corresponding time phase. STF has been widely used in many fields, such as land-cover change detection [20], land surface temperature retrieval [21], and crop research at the field scale [22]. Specifically, the data pairs used for STF in this study were MODIS data and Landsat data. Landsat data have a high spatial resolution of 30 m, but with a long satellite revisit cycle of up to 16 days. In contrast, the maximum spatial resolution of MODIS data is only 250 m, but the temporal resolution is relatively high, and one scene image can be obtained for the same area every day.

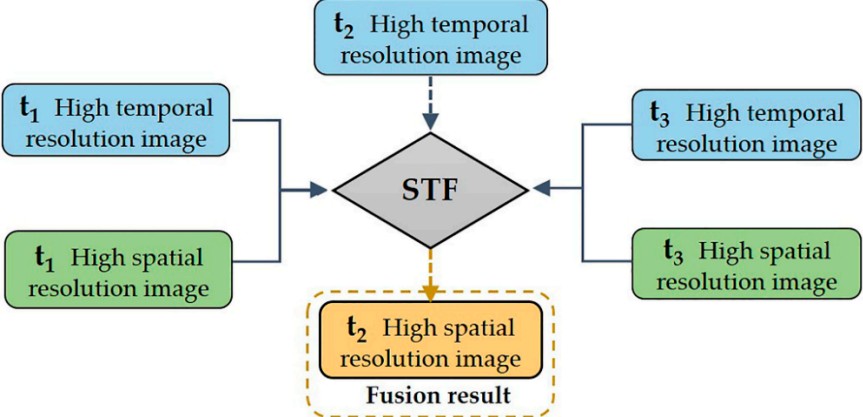

**Figure 3.** Schematic diagram of the spatiotemporal data fusion (STF) framework.

Among the many STF models, the enhanced spatiotemporal adaptive reflectance fusion model (ESTARFM) [19] is one of the outstanding methods. In this study, ESTARFM was used to supplement the remote sensing data of Taihu Lake, so as to ensure that we had no less than seven high-quality images in each season. This algorithm takes into account not only spectral information, but also pixel heterogeneity and spatiotemporal information, which greatly improves the prediction accuracy for large, heterogeneous regions. As for the details of ESTARFM, we refer the reader to the paper written by Zhu et al. [19].

Figure 4 shows the MODIS image acquired on 3 November 2014, two pairs of Landsat and MODIS images acquired on 30 July 2014 and 5 December 2014, respectively, the fusion result image, and the actual Landsat image acquired on 3 November 2014. From the perspective of visual analysis, the fusion result is very close to the actual Landsat image obtained on 3 November 2014, which indicates that the STF method successfully obtained these changes from MODIS observations to estimate the Landsat reflectance. To enhance the visual judgment, two detailed regions cropped from fusion result image and the actual Landsat image are shown in Figure 4b. It is clear that the distribution characteristics and cover area of the blooms in the simulated image are very close to the actual image. Although the fused result was slightly more blurred than the actual image, and also produced some errors, the fused images were still quite similar to the actual image. The above experiments show that the STF method is feasible and applicable for the generation of Landsat-like images at the required time.

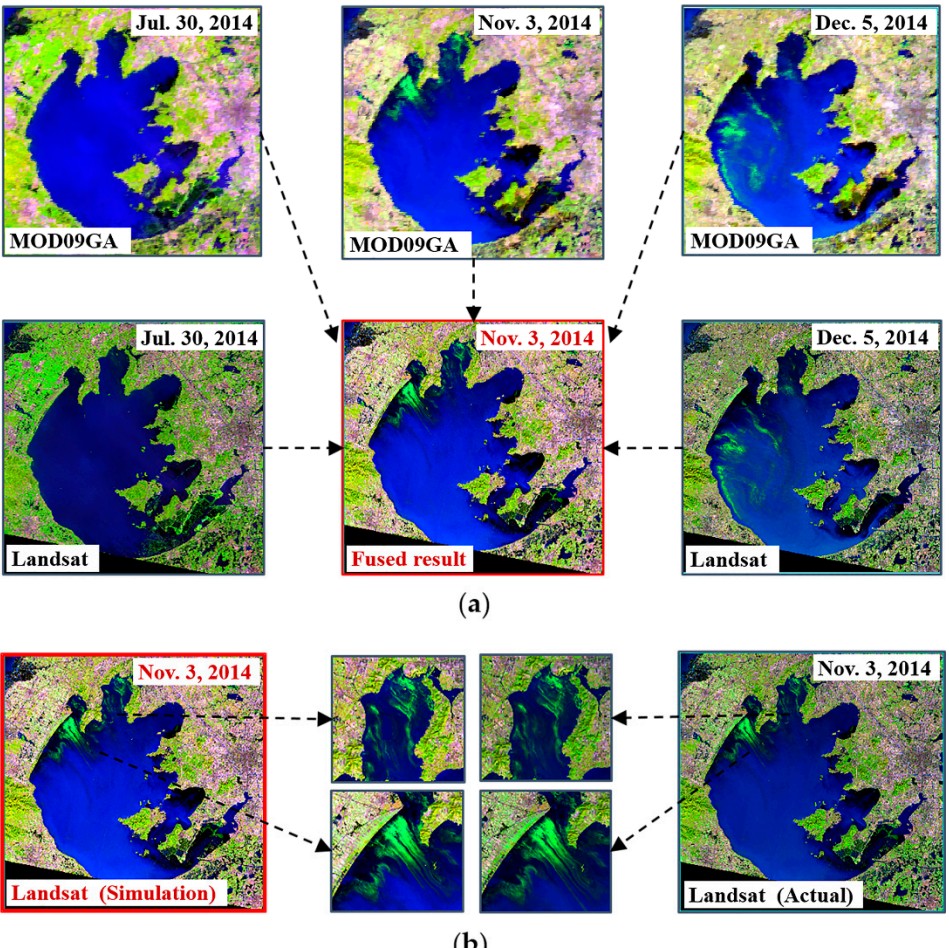

**Figure 4.** Simulated validation of the STF method (the order of the band: short-wave infrared, near-infrared, red). (**a**) Schematic diagram of the fusion of high-resolution images of Taihu Lake. (**b**) Comparison and validation between the simulated Landsat-like image obtained using the STF method and the actual Landsat image.

## 2.4. Algal Bloom Information Extraction from Remote Sensing Images

This paper is focused on the study of the temporal and spatial distributions of the algal blooms on the lake's surface rather than the total algae stock of the whole water. This makes remote sensing images a good data source for the study.

### 2.4.1. Algal Bloom Information Extraction from Optical Images

Different ground objects have different reflection characteristics. Li et al. [23] conducted two experiments at Taihu Lake in July and October 2006, where they measured the reflectance spectra of typical floating leaf plants, submerged plants, algal blooms, and the water bodies. It can be seen from Figure 5 that the spectral characteristics of the blooms are closer to those of aquatic plants (floating plants) than ordinary water bodies, with a reflection peak around 550 nm and a steep slope effect similar to that of the vegetation spectral curve features at 675–710 nm. In the near-infrared band (710–900 nm), algae cells have a very high reflectivity platform that is similar to the spectral characteristics of vegetation, due to the strong reflection of the solar spectral energy, while the reflectance spectrum of water rapidly decreases in the near-infrared band. This feature provides a theoretical basis for monitoring algal blooms by remote sensing.

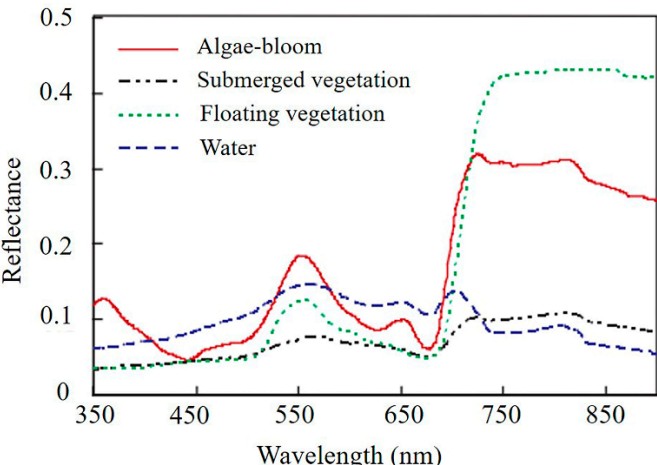

**Figure 5.** Representative reflectance spectra of the algae-bloom, water, submerged vegetation, and floating vegetation at Taihu Lake (from Li et al. [23]).

In general, the extraction methods for algal blooms in optical remote sensing imagery can be categorized into index-based threshold division methods and image-classification-based interpretation methods. The commonly used indices for algal bloom monitoring are the enhanced vegetation index (EVI), the normalized difference vegetation index (NDVI), and the floating algae index (FAI) [24–26]. Although the index threshold based methods can identify typical algal blooms, in many cases, they are affected by external disturbance such as the degree of turbidity of the water, the degree of cyanobacteria enrichment, and the atmospheric conditions (aerosols and atmospheric scattering). These factors make it difficult to unify the threshold for algal blooms extraction, which can lead to the relatively low reliability of extracting the information of low-concentration cyanobacteria blooms.

Among the many remote sensing image classification algorithms, the support vector machine (SVM) [27] method has a pleasing classification performance and has been used in many remote sensing applications [28–30]. This method can automatically find support vectors that have a greater discriminative ability for classification based on the principle of structural risk minimization, so as to maximize the interval between the hyperplane and the different sample sets. Therefore, SVM has a better generalization ability and can obtain a higher classification accuracy than many of the traditional classification algorithms. Figure 6 shows the Landsat-7 image acquired on 24 September 2011 and its supervised classification results for the algal bloom obtained by SVM. To enhance the visual judgment, two detailed regions cropped from Figure 6a,b are shown in Figure 6. It can be clearly seen that SVM has good classification performance in both the high- and low-concentration bloom areas.

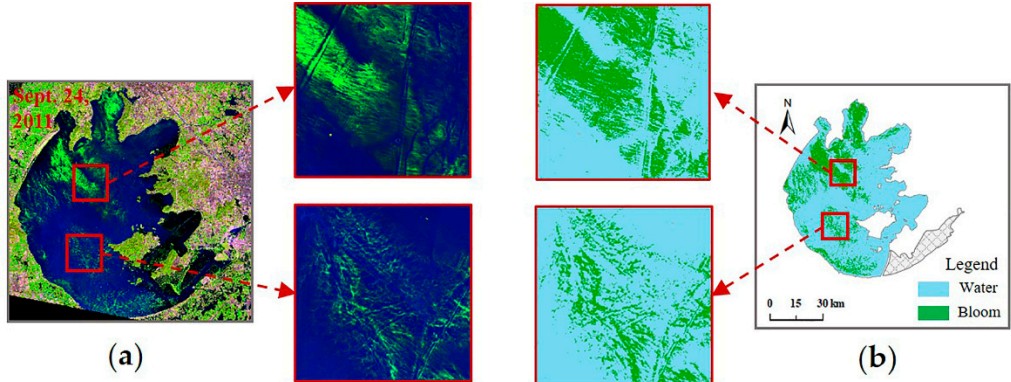

**Figure 6.** Validation of the support vector machine (SVM) method used to detect the algal blooms. (**a**) Landsat-7 image acquired on 24 September 2011. (**b**) Supervised classification results for the algal bloom obtained by SVM.

SVM was thus used in this research to extract the algal blooms from the optical images (Landsat 5–8, GF-1, and Sentinel-2) of the study area. Specifically, due to the special spectral characteristics of vegetation, we first masked the study area and used false-color synthesis to highlight the blooms in the lake. We then selected two types of samples—'algal blooms' and 'others'—and finally carried out SVM supervised classification to extract the blooms. When selecting training samples for the detection of algal blooms, the basic principle is that the samples must be representative, statistical, and accurate; and there must be at least 50 training samples of each type. Furthermore, we select training samples based on the characteristic spectral curves and pixel values.

### 2.4.2. Extracting Algal Blooms from Synthetic Aperture Radar (SAR) Images

Synthetic aperture radar (SAR) measures the polarization scattering characteristics of target objects by transmitting and receiving electromagnetic waves of different polarization channels and recording them in the form of a scattering matrix. The sentinel-1 data used in this study provide dual-polarization information, and each pixel data can be expressed by a polarization covariance matrix as follows:

$$C_2 = \begin{bmatrix} |S_{HH}|^2 & \sqrt{2}S_{HH}S^*_{HV} \\ \sqrt{2}S_{HV}S^*_{HH} & 2|S_{HV}|^2 \end{bmatrix} \tag{1}$$

where * is the conjugate transpose operator, and $S_{HV}$ denotes a radar wave of the target, transmitted in a vertical polarization form and received in a horizontal polarization form, which consists of the amplitude $|S_{HV}|$ and the phase $S_{HV} = |S_{HV}|e^{i\varphi_{HV}}$. The other elements are similarly defined.

In this study, the Wishart supervised classification algorithm [31] was used to classify the dual-polarization Sentinel-1 images and extract the algal blooms. The Wishart classifier is a classifier based on the maximum likelihood method, and its basic process is as follows. Firstly, some pixels of each object type are selected manually in the image, and we then calculate the average covariance matrix of each type and take it as the cluster center $C$. Then, for each target pixel, the Wishart distance between it and each cluster center is calculated. Finally, the pixel is classified as the type of ground object with the smallest Wishart distance. The Wishart distance of the polarization covariance matrix $Z$ of a pixel to the center $C_m$ of the class $m$ is derived as:

$$D(Z, \omega_m) = ln|C_m| + Tr\left(C_m^{-1}Z\right) \tag{2}$$

where $Tr$ denotes the trace of the matrix. The bloom has a certain thickness and viscosity, which can smooth the water surface waves. Furthermore, the biological surfactants released by algae can reduce the surface tension of water and reduce the backscattering of radar waves, thereby forming dark spots area on the radar image. Based on this feature, we used the Wishart supervised classification algorithm to categorize Taihu Lake into two object types: algal blooms and others. The classification results for a Sentinel-1 image are shown in Figure 7.

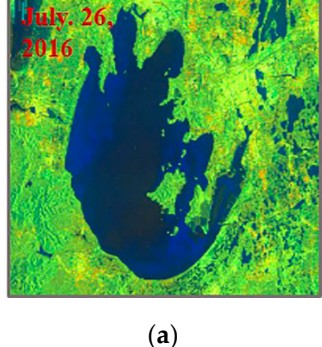

(**a**)

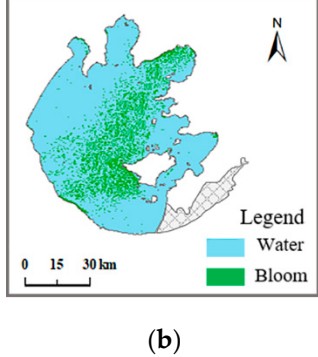

(**b**)

**Figure 7.** Example of algal bloom extraction in a SAR image. (**a**) Sentinel-1 image acquired on 26 July 2016. (**b**) The spatial distribution of the algal bloom obtained by the Wishart classifier.

## 3. Results and Discussion

### 3.1. Temporal and Spatial Distribution of Algal Blooms in Taihu Lake

According to the information extracted from the multi-source satellite remote sensing data from 2008 to 2017, we analyzed the spatial and temporal distributions of the algal blooms in Taihu Lake by three statistics: the average seasonal/annual coverage area, the hazard degree, and the spatial distribution of the bloom outbreak frequency.

The seasonal/annual average coverage area is the ratio of the total coverage area of the algal blooms in each season/year to the number of images in the corresponding season/year. The unit is km$^2$.

According to the study of Liu et al. [32], the hazard degree for algal blooms in Taihu Lake can be categorized into five degrees, which are shown in Table 2.

**Table 2.** Grading system for the algal bloom hazard degree in Taihu Lake.

| Element | Small | Medium | Large | Heavy | Extra Large |
|---|---|---|---|---|---|
| Bloom coverage area (km$^2$) | (0, 150] | (150, 400] | (400, 600] | (600, 900] | >900 |

The outbreak frequency of the algal bloom of pixel *i* is defined as:

$$F_i = N_i/T \tag{3}$$

where $N_i$ is the outbreak times of the algal blooms, and *T* is the total number of satellite images during a period. The larger the F value, the higher the outbreak frequency of algal blooms in this region.

#### 3.1.1. Average Algal Bloom Coverage Area

By recording the average seasonal/annual algal bloom coverage area for Taihu Lake in 2008–2017 (Figure 8), it can be seen that the coverage area for the algal blooms in spring, summer, and winter generally showed a downward trend during 2008–2015. After 2015, the coverage area gradually increased. The area in autumn showed a rising trend during 2008–2017, with small peaks in 2011 and 2015 and a maximum in 2017.

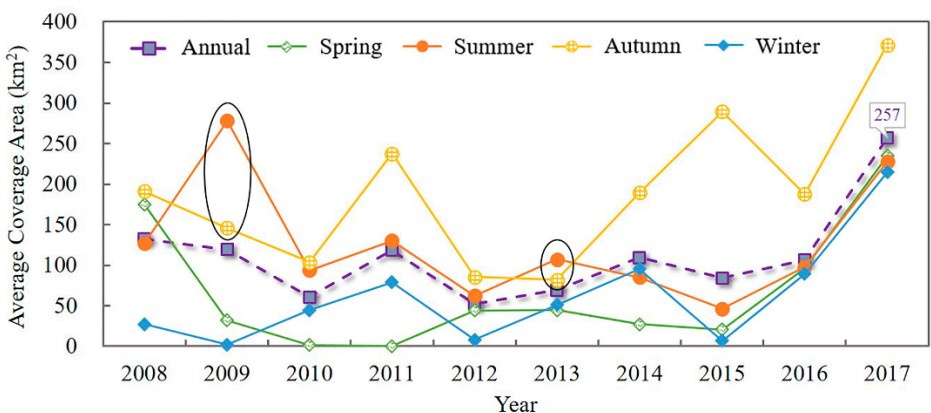

**Figure 8.** Average seasonal/annual algal bloom coverage area for Taihu Lake during 2008–2017.

Generally speaking, the coverage area of algal blooms in summer and autumn is higher than that of spring and winter. Moreover, in most years, the coverage area in autumn is significantly higher than that in the other three seasons. This indicates that, in the four seasons from 2008 to 2017, the average algal bloom coverage in autumn was the most serious, followed by summer. Taking a year as a unit, the coverage area of cyanobacteria blooms in Taihu Lake also showed a downward trend before 2015, rebounding after 2015, and reaching a maximum of 257 km$^2$ in 2017.

### 3.1.2. Variation Trend of the Algal Bloom Coverage Areas in Different Hazard Degrees

In order to explore the changes of the occurrence frequency of algal blooms in the different seasons at Taihu Lake, we recorded the proportion of images with different hazard degrees in each season of the first five years and the latter five years (Figure 9).

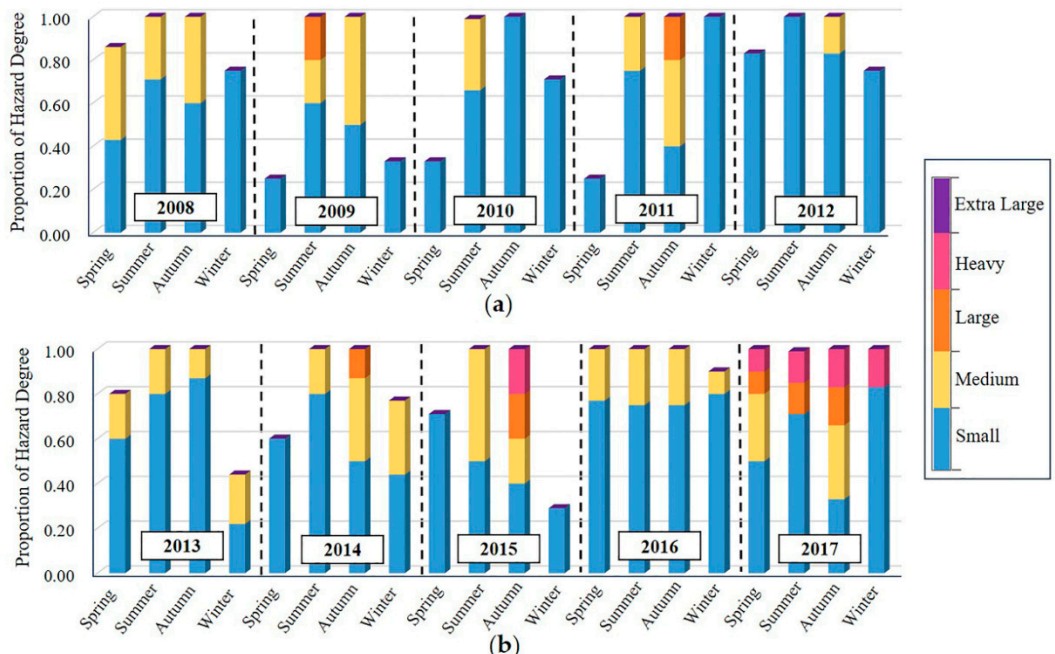

**Figure 9.** Proportion of the seasonal algal bloom coverage area of different hazard degrees in Taihu Lake. (**a**) Proportion of the seasonal algal bloom coverage area of different hazard degrees from 2008 to 2012. (**b**) Proportion of the seasonal algal bloom coverage area of different hazard degrees from 2013 to 2017.

Figure 9a shows that, between 2008 and 2012, algal blooms of medium and above hazard degrees mainly occurred in summer and autumn, suggesting that the outbreak of algal blooms is closely related to climatic conditions. As to the frequency of algal bloom outbreaks in each season (i.e., the height of each bar in Figure 9), the occurrence of algal blooms is very high in summer and autumn, and is much lower in spring and winter. All the above results imply that the algal blooms are most vigorous in summer and autumn. For the different years of the first five years, the outbreaks of algal blooms in the summer of 2009 and the autumn of 2011 were more serious. In addition, in 2008 and 2012, the blooms had a higher frequency of outbreaks.

As can be observed from Figure 9b, the occurrence frequency of blooms with medium and above hazard levels increased significantly after 2013, which indicates that the ecological environment and climate of Taihu Lake region changed to some degree in the latter five years. Similar to the previous five years, the serious algal blooms hazard degrees were still concentrated in summer and autumn. Furthermore, each season of 2016 and 2017 had a high frequency of algal bloom outbreaks. In 2017, algal blooms were detected from almost all the images, and there were 'heavy' algal bloom hazards in each season.

### 3.1.3. Spatial Distribution of the Occurrence Frequency of Algal Blooms

To further grasp the spatial distribution traits of algal bloom outbreaks in Taihu Lake from 2008 to 2017, outbreak frequency maps of each season (Figure 10a) and year (Figure 10b) were produced. It can be seen from Figure 10a that, from 2008 to 2012, the high-frequency areas of algal bloom outbreaks in spring mainly occurred in the western coast of Taihu Lake; in summer, the outbreak frequency of

algal blooms in the northern bays clearly increased, while the blooms decreased in the Southwest Lake area; the frequency of algal blooms in the northern bays and Meiliang Bay reached a maximum in the autumn; and in winter, the bloom coverage and outbreak frequency were significantly lower than in autumn, but the spatial distribution of the bloom outbreaks was still dominated by the northern lake zones.

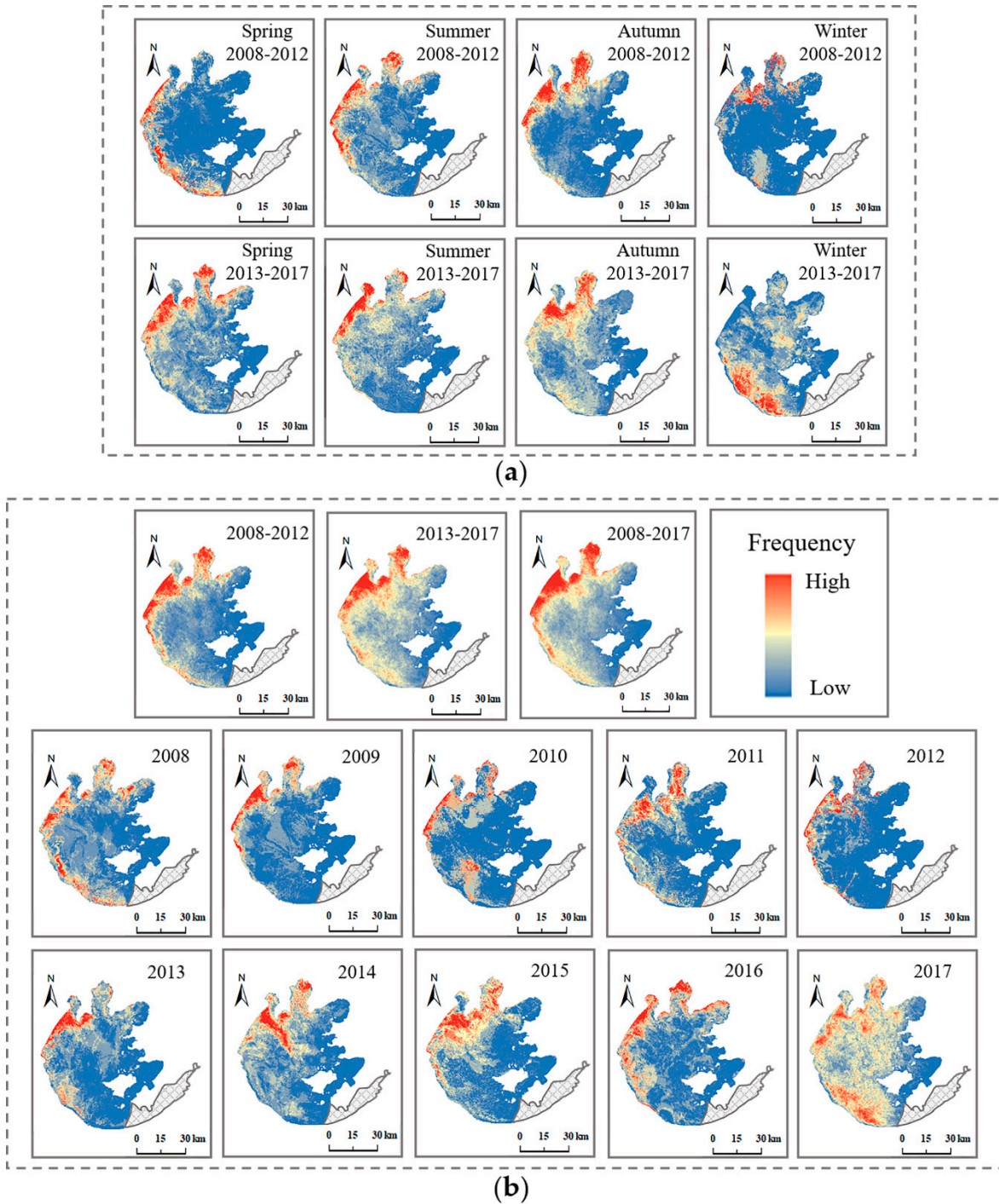

**Figure 10.** Spatial distribution of the outbreak frequency of algal blooms in Taihu Lake. (**a**) Spatial distribution of the algal bloom frequency in each season. (**b**) Spatial distribution of the bloom frequency in different years.

From 2013 to 2017, the spatial distribution of the high-frequency areas of algal bloom outbreaks in spring and winter was significantly different from that of the first five years, while the spatial distribution in summer and autumn was relatively similar. In addition, the outbreak frequency of algal blooms in the latter five years was generally higher than that in the first five years, and tended to expand toward the lake center.

Figure 10b shows that the high-frequency areas of algal bloom outbreaks during 2008–2012 were mainly distributed in the Northwest Lake area and the northern bays area. Compared with the first five years, the high-frequency areas of algal bloom outbreaks during 2013–3017 had a tendency to spread to the Central Lake area. For the different regions of Taihu Lake, the outbreak frequency of algal blooms in the Northwest Lake area and Meiliang Bay was the highest, followed by Zhushan Bay, Gong Bay, the Central Lake area, and the Southwest Lake area, and the outbreak frequency of blooms in the East Lake area was the lowest. A main cause of the spatial distribution traits of blooms in Taihu Lake is that, in the upper reaches of Taihu Lake, a large volume of industrial and domestic pollutants are discharged, which increases the concentration of nutrient salts in the water, resulting in the Northwest Lake area and the northern bays being the areas with a high-frequency of cyanobacteria blooms. Moreover, the water surrounding the bay areas has little fluidity, and compared with the Central Lake area, it is more affected by human activities, and the pollutants are more likely to accumulate, resulting in high eutrophication.

### 3.2. Environmental Driving Forces

Algae are sensitive to changes in environmental conditions, and the algae growth in Taihu Lake is influenced by both algal physiology and external factors [33]. In this section, to discuss the main environmental drivers of the algal blooms of Taihu Lake, the relationships between the environmental factors and the outbreaks of blooms are investigated.

### 3.2.1. Water Quality Analysis

Eutrophication is the main prerequisite for the outbreak of algal blooms, in which nitrogen and (especially) phosphorus have been considered to be the major nutrient elements [34]. The US Environmental Protection Agency (USEPA) pointed out in its "Guidelines for Lakes and Reservoirs Technical Guidelines - Nutrient Standards" that, in lakes and reservoirs, cyanobacteria bloom may occur when the total phosphorus (TP) and total nitrogen (TN) concentrations exceed 1 μg/L and 150 μg/L, respectively.

The annual average nitrogen and phosphorus concentrations in Taihu Lake from 2008 to 2017 were obtained through the Taihu Lake Basin Authority of the Ministry of Water Resources. As can be seen in Figure 11, during the study period, the concentrations of TN and TP exceeded the category IV and III standards for China's water bodies, respectively. Moreover, the lowest values of TP and TN concentrations are significantly higher than the above criteria, and they fully satisfy the nutrient concentration conditions required for the outbreak of the cyanobacteria bloom; therefore, the type of blooms that occurred during the study period was cyanobacteria blooms. Much more interestingly, the slope values of the annual variation of TN and TP are both low, which indicates that the concentrations of TN and TP have showed no obvious fluctuation during the decade, and are relatively stable.

Physiological characteristics, nutrients, and meteorological conditions all play key roles in the outbreak of algal blooms. Through the above analysis of the main water quality indicators, it is clear that the nutrient concentrations in Taihu Lake reached and broke the threshold for algal blooms, and were quite stable over the study period. That is to say, the nitrogen and phosphorus concentrations are no longer the main factors that resulted in the notable temporal changes of the distribution of blooms in Taihu Lake. Therefore, it is necessary to discuss and analyze the meteorological conditions.

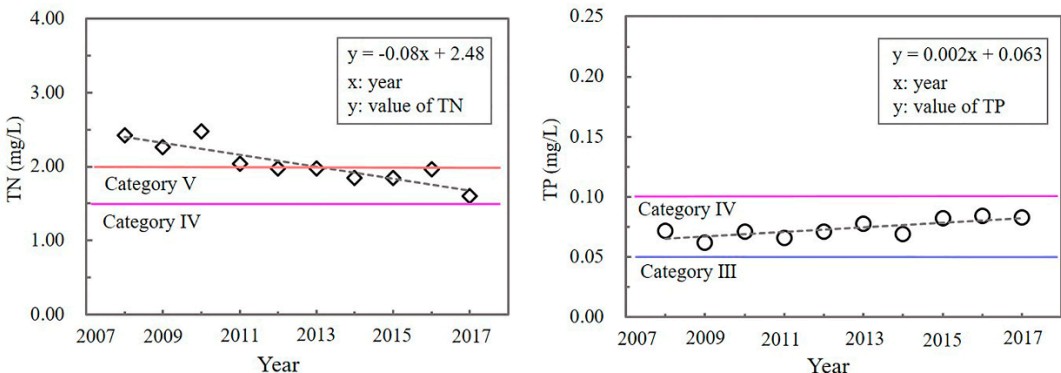

**Figure 11.** Different water quality indicators for Taihu Lake from 2008 to 2017.

### 3.2.2. Quantitative Analysis of the Meteorological Factors

In this study, we obtained the meteorological data for the two sites of Dongshan Station and Wuxi Station (as shown in Figure 1) for 2008–2017 from the National Meteorological Observatory. The metrological data included temperature (TEM), sunshine duration (SSD), precipitation (PRE), air pressure (PRS), and wind speed (WIN).

From the previous analyses on the temporal and spatial distribution of algal blooms in this study, it can be seen that, in the four seasons of 2008–2017, the bloom in autumn is the most serious; in winter, the algal blooms get worse in the latter years. In order to explore the impact of these meteorological factors on the seasonal variation of algal blooms, a Pearson correlation analysis was conducted between the algal bloom area and the average temperature, sunshine duration, wind speed, precipitation, and atmospheric pressure. The correlation values for the autumn and winter from 2008 to 2017 are displayed in Table 3.

**Table 3.** Pearson correlation coefficients between the meteorological variables and the algal bloom area from 2008 to 2017.

| Items | TEM/°C | | SSD/h | | WIN/m·s⁻¹ | | PRE/mm | | PRS/Pa | |
|---|---|---|---|---|---|---|---|---|---|---|
| | Autumn | Winter | Autumn | Winter | Autumn | Winter | Autumn | Winter | Autumn | Winter |
| Correlation coefficient | 0.186 | 0.380 | −0.723 * | 0.510 | −0.510 | −0.801 * | 0.361 | −0.757 * | 0.350 | 0.796 * |

* denotes a significance level of less than 0.05.

The results show that the length of sunshine duration in autumn is significantly negatively correlated with the algal bloom coverage; that is, in autumn, the sunshine duration has a significant inhibitory effect on the algal blooms. In winter, wind speed and precipitation variables are significantly negatively correlated with algal bloom coverage, while the pressure variable is significantly positively correlated with the algal bloom coverage. In other words, in winter, the lower the wind speed, the less the precipitation, and, the higher the air pressure, the larger the algal bloom coverage.

### 3.2.3. The Influence of Wind Direction on the Spatial Distribution of Algal Blooms

As revealed in Section 3.2, human activities and the geographical location have an influence on the spatial distribution of blooms. In fact, the spatial distribution of the algal bloom outbreak frequency in Taihu Lake is also related to the wind direction. To visualize this issue, we take 2017 as an example, and display the frequency chart of wind direction in each season for the Taihu Lake region in Figure 12.

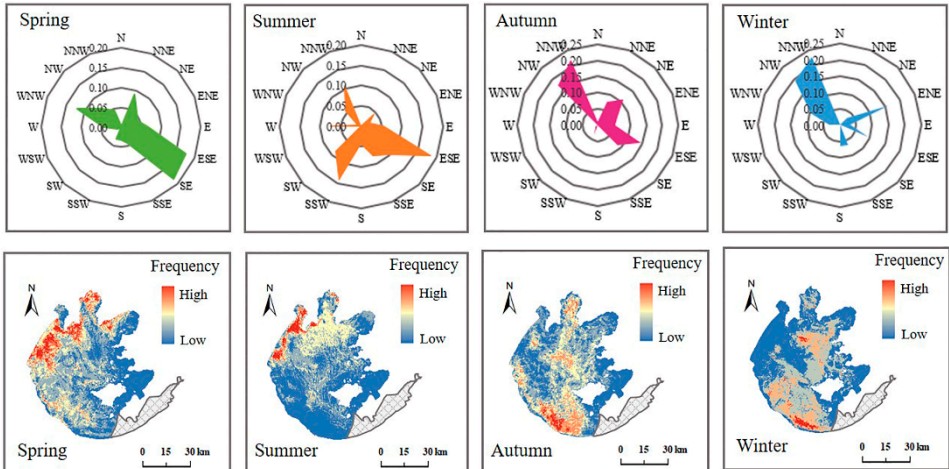

**Figure 12.** The wind direction rose charts and the spatial distributions of algal bloom frequency in the four seasons of 2017 for Taihu Lake.

It can be seen from Figure 12 that, during the recovery period of cyanobacteria blooms in the spring of 2017, SE and ESE direction winds prevailed in the Taihu Lake region, and the blooms gathered and drifted to the northwest, due to the wind, thus forming a high-frequency area in the Northwest Lake area. During the summer, when algal blooms are active, the dominant wind direction was still ESE, accumulating the bloom drifting from the southeast and the cyanobacteria originally growing in the northwest, resulting in serious algal bloom in the Northwest Lake area, Zhushan Bay, and Meiliang Bay.

In autumn, the algal blooms are still active, but the dominant wind direction turns to the north. Under the impact of the relatively high-frequency NNW, NW and ESE winds, a large number of blooms occurred in the south, and the frequent recurrence area shifted to the Central Lake and Southwest Lake areas. In winter, under the influence of the NNW, NW, and ENE winds, the high-frequency zone of algal blooms continued to transfer to the Central Lake and Southwest Lake areas. Therefore, wind direction is an important driving factor affecting the distribution pattern of algal blooms in Taihu Lake.

## 4. Conclusions

Satellite remote sensing technology provides us with the possibility to quickly and extensively monitor algal blooms in water. However, limited by the satellite sensor revisit cycle and weather conditions, high-quality remote sensing images cannot be obtained in certain time periods, resulting in inadequate algal bloom information and an inability to comprehensively reflect the long-term and fine-scale development trends of algal bloom outbreaks. In this study, multi-source optical and radar remote sensing data from 2008 to 2017 were used to identify and extract the algal bloom information of Taihu Lake, and images with a high spatial resolution generated by STF were used as supplementary data. On the basis of the comprehensive analyses of the temporal and spatial distributions of the algal blooms, the water quality and weather data for Taihu Lake were employed to investigate the main environmental drivers of algal blooms. The major conclusions of this study are as follows:

1.  From 2008 to 2017, the outbreak coverage area of algal blooms in autumn was the most serious, followed by summer. Before 2015, the average annual coverage of algal blooms was stable, but this fluctuated significantly after 2015. Compared with the first five years, the frequency of blooms of the medium and above hazard degrees increased significantly between 2013 and 2017.

2.  From the perspective of the seasonal scale, the spatial distribution of the algal blooms in the summer and autumn of the latter five years was basically the same as that of the first five years, and the algal blooms were mainly concentrated in the Northwest Lake area, Meiliang Bay, and Zhushan Bay. However, the spatial distribution of the high-frequency areas of spring and winter

algal bloom outbreaks was significantly different in the first and the second five years, especially in the southern and northern parts of the lake, where there was a relatively significant migration and transformation. On the yearly scale, the high-frequency areas of Taihu Lake bloom outbreaks in the latter five years tended to expand toward the lake center, and the algal bloom outbreaks in the Northwest Lake area and the northern bays were still relatively high.

3.  During the study period, both the nitrogen and phosphorus concentrations in Taihu Lake met the threshold values for algal bloom outbreaks and were relatively stable. According to the analysis of meteorological factors and algal bloom outbreaks, the length of sunshine in autumn can significantly inhibit algal blooms. Moreover, the area of algal blooms in Taihu Lake has notable correlations with the sunshine duration, wind speed and direction, precipitation, and air pressure.

**Author Contributions:** Conceptualization: T.Z., H.H. and X.M.; Methodology: T.Z., H.H. and X.M.; Software: T.Z.; Validation: T.Z., H.H. and X.M.; Formal analysis: T.Z.; Investigation: T.Z.; Resources: T.Z., H.H., X.M. and Y.Z.; Data curation: T.Z.; Writing—Original draft preparation: T.Z.; Writing—Review and editing: T.Z., H.H. and X.M.; Visualization: T.Z. and X.M.; Supervision: H.H., X.M. and Y.Z.; Project administration: T.Z., H.H., X.M. and Y.Z.; Funding acquisition: H.H. All authors have read and agreed to the published version of the manuscript.

**Funding:** This research was funded by the National Natural Science Foundation of China under grant 41701390 and 41704036.

**Acknowledgments:** The authors acknowledge Anhui Provincial Key Laboratory of Wetland Ecosystem Ecological Protection and Restoration for their support. We would like to thank the data providers of the USGS, ESA, NASA, CNSA-GEO, and the National Meteorological Information Center. We would also like to thank Xiaolin Zhu (Hong Kong Polytechnic University) for providing access to the ESTARFM IDL code.

**Conflicts of Interest:** The authors declare no conflict of interest.

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
