# Peer review of "Long-Term Spatiotemporal Variation and Environmental Driving Forces Analyses of Algal Blooms in Taihu Lake Based on Multi-Source Satellite and Land Observations"

_water, doi:10.3390/w12041035_

Round 1

Reviewer 1 Report

Dear editor, authors

I have reviewed the manuscript entitled 'Long-term Spatiotemporal Variation and Environmental Driving Forces Analysis of Algal Blooms in Taihu Lake Based on Multi-Source Satellite and Land Observations from Zhang et al. submitted to Water. The authors describe the occurrence of algal blooms as detected by optical and radar satellite sensors using a SVM classification approach. By collecting a long time series of satellite data (2008 to 2017) they describe the algal bloom dynamics throughout different seasons and link it to potential driving forces such as nutrient concentrations, temperature, sunshine duration, wind speed, precipitation and air pressure.

To achieve their goal the authors perform a significant amount of satellite data processing such as all the necessary preprocessing of multiple optical and radar data sets. To remove technical issues such as missing data they also apply a data fusion approach (TSF) enabling them to fill up data gaps due to cloud cover and thus missing data.

I congratulate the authors with their work as it combines many different aspect of remote sensing for algal bloom detection and I agree with them that satellite observations offer powerful tools to better understand the ecosystem. Which they also demonstrate. Still, some crucial information is missing before I can conclude that the method presented is fit for purpose. Because of this I suggest a major review is needed before this work can be published.

The following information is missing:

1) Which water types are present in the lake? Can you see the bottom (1.9m water depth)?

2) linked to 1): Which atmospheric correction procedures were used for the satellite data? How accurate are the L2 products and which ones are exactly used in the SVM?

3) Some sort of validation of the SVM method used to detect the algal blooms. How accurate is it?

4) How do classification results from optical sensors vs. radar sensors compare. Do they result in similar products or are there differences. This can be done by comparing algal bloom detections by the different sensors on the same day.

5) How well does the TSF approach works in your use case? No validation/intercomparison is provided.

As satellite data is the key source of data for your study, any inaccuracies or quality issues in this data will propagate through further analysis. Example, the authors notice an increase in algal blooms from 2015 (figure 8) but that is also the period that the Sentinel sensors were introduces into the study so I cannot determine if there is actually a real increase in blooms are there is a shift cause by changing data types. For example, we notice significant sun glint effects in S2 data on water surfaces. Could this be mis-classified as algal blooms? Is there a correction for sun glint in the atmospheric correction?

Other suggestions/comments:

1) Please introduce sooner: which sensors you are actually using

2) Describe clearly how you describe an algal bloom. When is an algal concentration considered a bloom? (E.g. L128-129, vague description of algal bloom 'information' and 'quality indicators')

3) Describe what type of algal blooms are present in Lake Taihu? I guess that you are focusing on floating blooms? Are there also algae present in the water column and are they detected? Can you differentiate between them?

4) Fig 4: please make figures bigger, red text on figure is very difficult to read

5) Fig 7: it is difficult to see how SAR sees an algal bloom (image on the left). What should the reader look for?

6) Fig 1O: change date range from 2008-2012 to 2013-2017 in second row (summer, autumn, winter)

Best regards

Reviewer 2 Report

In this study, the authors have looked at the development of cyanobacteria blooms of public health as well as ecosystematic concerns in a lake in China. The study used Multi-source satellite and onsite observations of several factors on a long-term scale to address the problematic expanding  CHAB. The study is well designed and introduced and the M&M are described adequately and the results and conclusions are in line. The results of this study should be available for further research on CHAB development and the factors that are contributing to their intensification worldwide, notably the coupling of Remote-sensing with field measurement. Aside for a few structural and language editing, the manuscript should be considered for publication, pending remarks and comments from other reviewers align. 

Round 2

Reviewer 1 Report

Dear Authors, 

congratulations with your work. I consider the adaptation of your manuscript adequate as it deals with the concerns that I had.

I consider this work fit for publication in Remote Sensing.

Have a nice day